# DMRIntTk: Integrating different DMR sets based on density peak clustering

**Wenjin Zhang[1], Wenlong Jie[2], Wanxin Cui[2], Guihua Duan[2], You Zou[2,3]\*,
Xiaoqing Peng[1]\***

**1** Center for Medical Genetics & Hunan Key Laboratory of Medical Genetics, School of Life Sciences, Central South University, Changsha, China, **2** Hunan Key Laboratory of Bioinformatics, School of Computer Science and Engineering, Central South University, Changsha, China, **3** High Performance Computing Center, Central South University, Changsha, China

\* zouyou@csu.edu.cn (YZ); xqpeng@csu.edu.cn (XP)

**Data Availability Statement:** The methylation array datasets involved in this paper were downloaded from Gene Express Omnibus (GEO) with accession ids GSE48472, SE50192, GSE112047, GSE157272, GSE43414, GSE105109,

## Abstract

### Background

Identifying differentially methylated regions (DMRs) is a basic task in DNA methylation analysis. However, due to the different strategies adopted, different DMR sets will be predicted on the same dataset, which poses a challenge in selecting a reliable and comprehensive DMR set for downstream analysis.

### Results

Here, we develop DMRIntTk, a toolkit for integrating DMR sets predicted by different methods on a same dataset. In DMRIntTk, the genome is segmented into bins, and the reliability of each DMR set at different methylation thresholds is evaluated. Then, the bins are weighted based on the covered DMR sets and integrated into final DMRs using a density peak clustering algorithm. To demonstrate the practicality of DMRIntTk, it was applied to different scenarios, including tissues with relatively large methylation differences, cancer tissues versus normal tissues with medium methylation differences, and disease tissues versus normal tissues with subtle methylation differences. Our results show that DMRIntTk can effectively trim regions with small methylation differences from the original DMR sets and thereby enriching the proportion of DMRs with larger methylation differences. In addition, the overlap analysis suggests that the integrated DMR sets are quite comprehensive, and functional analyses indicate the integrated disease-related DMRs are significantly enriched in biological pathways associated with the pathological mechanisms of the diseases. A comparative analysis of the integrated DMR set versus each original DMR set further highlights the superiority of DMRIntTk, demonstrating the unique biological insights it can provide.

### Conclusions

Conclusively, DMRIntTk can help researchers obtain a reliable and comprehensive DMR set from many prediction methods.

GSE125895, GSE43414, GSE66351, and
GSE80970, and from the Religious Orders Study
and Memory and Aging Project (ROSMAP) cohort
[39] with Synapse ID syn3157275.

**Funding:** This work was supported in part by the
Natural Science Foundation of Hunan Province
(No. 2022JJ30694 and No. 2022JJ30750); Central
South University Innovation-Driven Research
Programme (No. 2023CXQD065); Special Funds
for Construction of Innovative Provinces in Hunan
Province (NO. 2023GK1010). The funders had no
role in study design, data collection and analysis,
decision to publish, or preparation of the
manuscript.

**Competing interests:** The authors have declared
that no competing interests exist.

# 1 Background

DNA methylation is a crucial epigenetic modification that plays a pivotal role in various biological processes, such as tissue differentiation, embryonic development, tumorigenesis and aging. Differentially methylated regions (DMRs) are genomic regions that display significant variations in DNA methylation levels between different biological states (e.g., normal versus diseased). Since these regions have been implicated in the regulation of gene expression and are increasingly recognized for their associations with a variety of diseases and biological processes, identifying DMRs is a fundamental task in DNA methylation analysis. The identified tissue-specific [1, 2], cell type-specific [3, 4] and disease-associated DMRs [5–7] can help to investigate the underlying molecular mechanisms of differentiation and pathogenesis, and serve as potential methylation biomarkers for screening diseases.

Many methods have been developed for detecting DMRs on methylation array data. These methods can be broadly categorized into CpG-based and candidate-region-based methods. Currently, CpG-site-based methods are very popular due to their fine resolution in detecting differential methylation at individual CpG sites. In CpG-site-based methods [8–13], differentially methylated CpGs (DMCs) are first identified based on the methylation levels of the samples between two groups. The p-values of DMCs are often corrected based on the auto-correlations of CpG site, the correlations between adjacent CpG sites, the p values of a adjacent CpG sites, etc. Then adjacent DMCs are agglomerated to form DMRs if some defined criteria are satisfied, such as a minimum distance between neighboring DMCs.

ProbeLasso [8] identifies DMRs by evaluating differential methylation at each CpG site using a linear regression model. It defines flexible, dynamic boundaries around each probe, extending upstream and downstream based on the genomic feature type the probe is located in. A region is selected if the number of significant probes within the Probe Lasso boundary exceeds a user-defined threshold. Finally, ProbeLasso computes a correlation matrix and applies Stouffer's method to calculate a P-value for each region, enabling the identification of significant DMRs. ProbeLasso can avoids bias toward probe-dense regions, enabling more comprehensive use of array data and highlight hypomethylated transcription factor binding motifs in potential novel pathways. However, both real data and simulation data have shown that ProbeLasso lacks sufficient power to identify the most significant DMRs when the effect size is small [14].

DMRcate [9] fits a linear model to each CpG site using empirical Bayes methods and applies Gaussian kernel smoothing to adjust the squared EWAS t-statistics at each CpG site, incorporating the squared t-statistics of adjacent CpG sites. The resulting smoothed statistics are then used to recalculate p-values for each CpG site. Significant CpG sites that are in close proximity to one another are subsequently grouped to form differentially methylated regions (DMRs). While DMRcate is computationally efficient, it does not account for correlations between neighboring sites, which has led to higher false positive rates in regions with strong inter-site correlations.

Comb-p [10], like DMRcate, combines EWAS summary statistics from neighboring CpG sites. It estimates spatial auto-correlation at various distance lags and adjusts p-values for each CpG site based on neighboring p-values. A peak-finding algorithm identifies regions with low p-value enrichment. The tool requires only the genomic location and p-value of the CpG as input, enabling DMR identification from meta-analyses or published statistics. The false discovery rate is then calculated, and significant regions are identified. While efficient, its sensitivity and specificity depend on the dataset used.

In contrast, candidate-region-based methods focus on larger genomic regions or predefined functional areas where methylation changes are likely to be biologically significant. In

candidate-region-based methods [15–25], there are typically two types of candidate regions: sample-independent regions and sample-dependent regions. The sample-independent regions are either predefined based on functional regions, such as CpG islands and shores, or generated by a sliding window on the genome. The sample-dependent regions are generated according to the characteristics of samples, including the coverage, depth of CpG sites, and methylation levels of CpG sites, or the methylation changes of CpG sites among multiple samples. Once the candidate regions are determined, DMRs are then identified by comparing the methylation levels of these regions across different samples.

Bumphunter [15] uses linear regression to model differential methylation between case and control groups at each CpG site. Candidate regions (bumps) are identified as clusters of consecutive probes with t-statistics above a user-defined threshold. Permutation tests are applied to estimate the statistical significance of these regions, separated by a minimum distance (maxgap). Spatial correlation models account for the methylation correlation between neighboring CpGs. While bumphunter effectively identifies biologically relevant epigenomic regions in microarray data, it has limitations, such as being unable to detect single base changes due to smoothing.

Seqlm [16] is a method for DMR identification that uses the minimum description length (MDL) principle to determine region boundaries, addressing model selection. It employs linear mixed models to assess the significance of these regions. Compared to other methods, Seqlm is more sensitive, specific, and efficient, requiring minimal parameter tuning and offering the fastest runtime. It divides the genome into segments based on probe distances and consistent methylation profiles, then tests for associations with phenotypes using a linear mixed model. However, Seqlm does not account for covariates, which limits its use in heterogeneous tissues where cell composition adjustments are needed.

Due to the different strategies used in different methods, different DMR sets are predicted on a same dataset, in terms of the DMR length, the number of probes and CpG sites included, and the methylation differences. There is no method that can perform well on datasets in all scenarios, and we can hardly figure out which scenario a method is most applicable to. Therefore, it is challenging to select a desirable DMR set for downstream analysis.

As we know, different methods have their own advantages in predicting different types of DMRs, and DMRs that are detected by most methods and with relatively large methylation differences can be regarded as highly reliable DMRs. Therefore, we develop a toolkit, DMRIntTk, which evaluates the reliability of different DMR sets and integrates them using a density peak clustering (DPC) algorithm. To evaluate the performance of DMRIntTk, it was applied to the DMR sets in four representative scenarios, including datasets with large methylation differences (five different tissues), medium methylation differences (the prostate cancer (PCa) tissues versus the adjacent normal prostate tissues, and the benign versus other five histological stages of PCa tissues), and small methylation differences (brain regions of patients with Alzheimer's disease (AD) versus the normal ones). The results show that DMRIntTk can enhance the proportion of DMRs with higher methylation differences. In addition, the overlap analysis suggests that the integrated DMR set is more comprehensive and reliable than individual DMR sets.

## 2 Methods

### 2.1 The pipeline of DMRIntTk

In this paper, a toolkit, DMRIntTk, is developed to integrate the DMR sets predicted by different methods. The pipeline of DMRIntTk mainly contains four steps, including constructing the reliability matrix, segmenting the genome, weighting the bins and integrating DMRs, as shown in Fig 1. The DMRIntTk begins by processing the pre-processed methylation array data

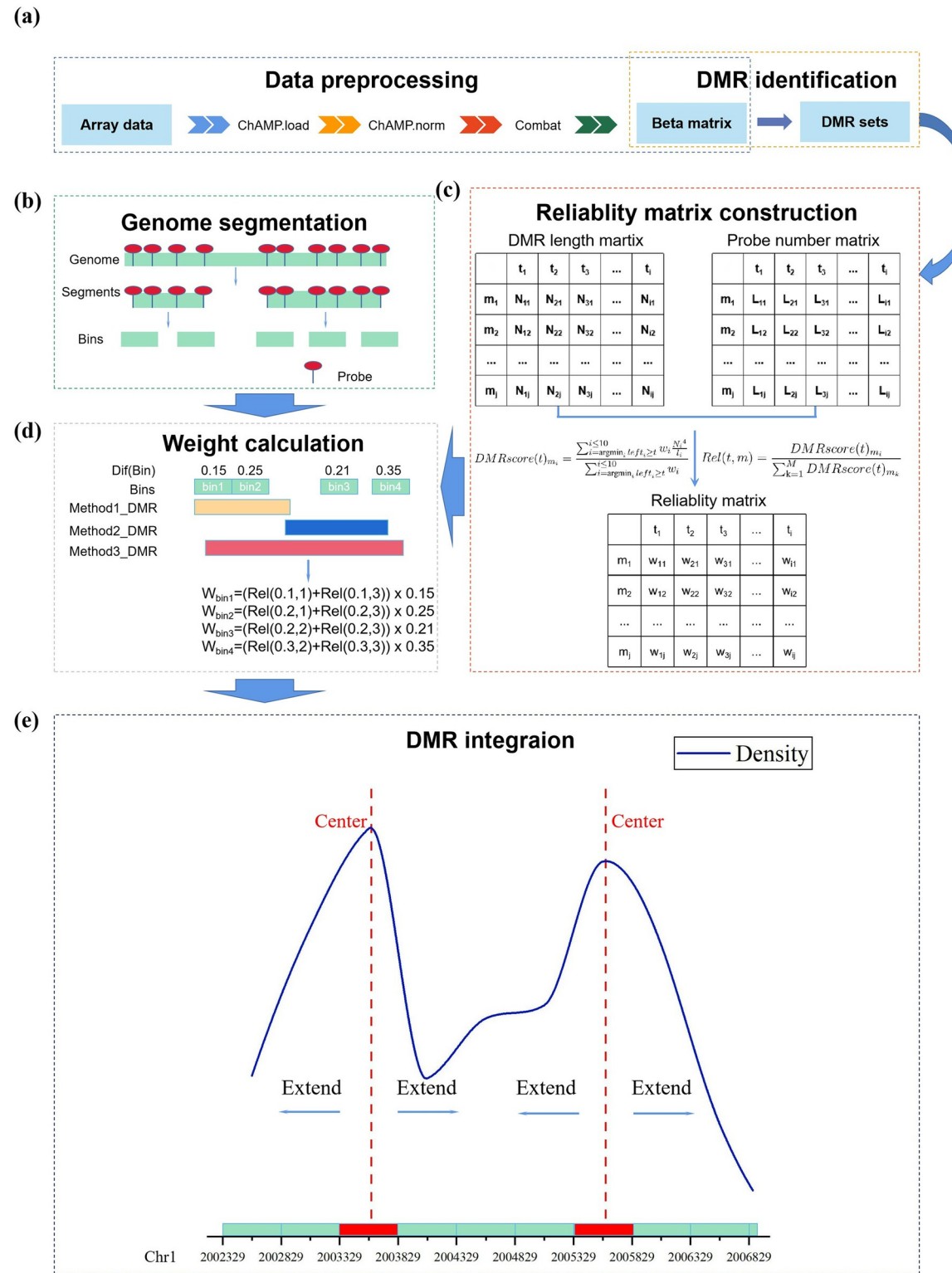

**Fig 1. A schematic diagram of DMRIntTk.** (a) Data pre-processing and DMR identication steps output DMR sets are used as standard input in DMRIntTk. There are four major parts in DMRIntTk, including (b) segmenting the genome, (c) constructing the reliability matrix, (d) weighting bins and (e) integrating bins based on density peak clustering. The example of each part are shown.

and applies various DMR detection methods to identify DMRs. For each identified original DMR set, its reliability matrix is constructed for each methylation difference threshold. Simultaneously, the genome is segmented into non-overlapping bins. Next, the weight of each bin is computed based on its methylation difference and corresponding reliability. Finally, an DPC algorithm, optimized for regional integration, is employed to integrate the bins, producing the final output: an integrated DMR set. Regarding inputs, the user simply needs to provide the pre-processed methylation array data and phenotype data, along with specified parameters, including one or more desired DMR prediction methods, array type, the minimum number of probes, and DMR preference (such as a preference for longer regions or regions with a higher probe density). The output is an integrated DMR set, which is provided as a file containing the final integrated DMRs, including the chromosome, start, and end position information for each DMR. The details of each step is described in following subsections.

## 2.2 Construct reliability matrix

Since different methods have their own preferences and advantages in predicting DMRs with different properties, the reliability of DMR sets predicted by different methods should be fully accessed. Therefore, we propose a metric *DMRscore* to evaluate the reliability of different DMR sets.

Let $m_{max}$ denote the maximum methylation difference of DMRs. Then, divide the range of methylation difference $[0, m_{max}]$ into ten intervals with equal length $m_{max}/10$. For each DMR set predicted by a method, the total number of probes and the total length of DMRs with methylation differences falling in each interval are calculated, respectively. Given a certain value of a threshold $t$ in the range of $[0, m_{max}]$, the *DMRscore* of a DMR set $m_k$ is calculated as Eq 1.

$$DMRscore(t)_{m_k} = \frac{\sum_{i=\underset{i}{\operatorname{argmin}}\, left_i \geq t}^{i \leq 10} w_i \frac{N_i^4}{l_i}}{\sum_{i=\underset{i}{\operatorname{argmin}}\, left_i \geq t}^{i \leq 10} w_i} \tag{1}$$

In Eq 1, $i$ denotes the $i$-th interval, where $left_i$ and $right_i$ denote the left and right boundaries points of the $i$-th interval, respectively. $w_i$ denotes the weight of the $i$-th interval, set as the average of the left and right boundaries, i.e., $\frac{left_i + right_i}{2}$. $N_i$ and $l_i$ denote the total number of probes and the total length of DMRs with methylation differences falling in the $i$-th interval, respectively. The term $N_i^4$ serves to emphasize the contribution of intervals with a larger number of probes by raising the probe count to the fourth power. This choice amplifies the influence of intervals with higher probe densities, reflecting their greater reliability in identifying significant methylation differences [26, 27].

With a certain threshold $t$, if the *DMRscore* of a particular DMR set is greater than that of other others, it indicates that the DMR set is more reliable and comprehensive with a higher probe density and more probes in DMRs whose methylation differences are greater than $t$. With $M$ DMR sets and a given threshold $t$, the normalized reliability score of each DMR set $m_k$, denoted as $Rel(t, m_k)$, can be calculated according to Eq 2.

$$Rel(t, m_k) = \frac{DMRscore(t)_{m_k}}{\sum_{n=1}^{|M|} DMRscore(t)_{m_n}} \tag{2}$$

The reliability scores of $M$ DMR sets can be calculated for each value of $t$ in the set $\{0, m_{max}/10, 2 * m_{max}/10, ..., 9 * m_{max}/10\}$. Then, a reliability matrix $R$ with $M \times 10$ can be constructed.

## 2.3 Segment the genome and weight the bins

For methylation array data, the whole genome is segmented into non-overlapping genomic bins based on the distances of probes. First, adjacent probes with a genomic distance of less than 500 bp were clustered together to form fragments. Then, a 500 bp sliding window was applied to segment fragments larger than 500 bp, dividing them into continuous, non-overlapping bins. For 450K array, 10,9524 bins are generated, and for 850K array, 17,9783 genomic bins are generated.

To obtain a reliable and comprehensive DMR set from the predictions of multi methods, each bin is evaluated by considering the methylation difference and the reliability scores of the covering DMR sets. For a bin $B_i$, let $Dif(B_i)$ denotes the absolute value of the methylation difference between two groups on $B_i$, and $N$ denotes the number of DMR sets covering $B_i$. Then, the weight of $B_i$ can be calculated by integrating the reliability scores of $N$ DMR sets and the methylation difference $Dif(B_i)$, as defined in Eq 3.

$$w(B_i) = \sum_{k=1}^{|N|} Rel(Dif(B_i), m_k) \times Dif(B_i) \tag{3}$$

## 2.4 Generate integrated DMRs based on density peak clustering

Bins covered by more DMR sets and with higher methylation differences will be assigned with higher weights, which are the basic elements in generating integrated DMRs. We applied an adapted density peak clustering (DPC) algorithm on weighted bins to identify DMRs. In this algorithm, the bins are treated as the basic points. The local density is defined as the weight of a bin, and the distance between two bin is defined as their genomic distance.

Assume that there two thresholds, $c_t$ and $n_t$, which are used for identifying cluster centers and cluster members, respectively. Firstly, bins with weights greater than $c_t$ are identified as cluster centers. Then, sequential adjacent cluster centers will be merged into one cluster center. For each bin which is not a cluster center and between two cluster centers, it will assigned to the cluster with a smaller genomic distance. Then pre-defined regions are formed by cluster centers and the assigned bins.

For each pre-defined region, a cluster will be identified by a cluster center extension mode. A cluster is initialized as the cluster center in a pre-defined region. Then, for the nearest bin in the pre-defined region from each side of the cluster, if its weight is greater than $n_t$, it will be merged into the cluster and the extension on this side will be continued until the weight of the next nearest bin is not greater than $n_t$. The finally formed cluster is considered as an integrated DMR and its methylation difference is calculated as the mean methylation difference of all clustered bins.

## 2.5 The DMR set integrated by DMRIntTk

Since there are two parameters in the DPC algorithm to identify DMRs, different settings of $c_t$ and $n_t$ will result in different integrated DMR sets. To facilitate the researchers and make the usage simple, DMRIntTk provides an automatic mechanism for determining the parameter values and outputting the final integrated DMR set.

Given that $m_{max}$ denote the maximum methylation difference of DMRs, the value of $c_t$ will be enumerated from the set of $\{0.2^*m_{max}, \ldots, 1^*m_{max}\}$ and the value of $n_t$ will be enumerated from the set of $\{0.1^*m_{max}, \ldots, 0.9^*m_{max}\}$. For each combination of $c_t$ and $n_t$, the DPC algorithm is applied to obtain an integrated DMR set. Then, for each integrated DMR set, the total number of contained probes, the total length of DMRs, and a proportion of DMRs with methylation differences greater than $0.5 * m_{max}$ are calculated. Based on these information, the final integrated DMR set output by DMRIntTk will satisfy the following two conditions: (1) contains a higher proportion of DMRs with methylation differences greater than $0.5 * m_{max}$ than any individual DMR set; (2) has the most probes or the longest total DMR length than other integrated DMR sets.

## 3 Results

In this study, seven state-of-the-art DMR detection methods, including Bumphunter [15], ProbeLasso [8], DMRcate [9], comb-p [10], ipDMR [11], mCSEA [28], and seqlm [16], were applied to predict DMRs on DNA methylation 450K array data in four scenarios. For the parameters in each method, the default values were adopted. For each DMR set, DMRs with adjusted p-values greater than 0.05 or containing less than 3 probes were filtered out. Finally, DMRIntTk was applied to integrate the original DMR sets predicted by seven methods.

To evaluate the integrated DMR sets obtained by DMRIntTk, the methylation difference distribution of the integrated DMR sets and the original DMR sets were compared. Further, an overlap analysis was carried between the integrated DMR sets and the original DMR sets. Moreover, the functional enrichment analysis of the integrated DMR set was analyzed and the impact of different genome segmentation strategies on integrated DMRs was evaluated.

### 3.1 Materials

In this paper, methylation array datasets of four scenarios are involved, including 1) five tissues, 2) the PCa tissues versus the adjacent normal prostate tissues, 3) the benign versus other histological stages of PCa tissues, and 4) the brain tissues with AD versus the normal brain tissues. The DMRIntTk is used to integrate DMR sets predicted by different method on the methylation array datasets in these scenarios.

In scenario 1, five tissues were involved, including 5 liver tissues, 6 muscle tissue and 6 omentum tissues extracted from GEO with accession id GSE48472 [29], and 5 lymphoid tissues and 7 tonsil tissues from GSE50192 [30, 31]. In scenario 2, the 450K methylation profiles of 31 PCa tissues and 16 adjacent normal prostate tissues were extracted from GSE112047 [32]. In scenario 3, the 450K methylation profiles of PCa tissues under different histological stages were extracted from GSE157272 [33], which contains 10 benign prostate tissues, 7 proliferative inflammatory atrophy (PIA) prostate tissues, 6 high grade prostatic intra-epithelial neoplasia (HGPIN) tissues, 7 Indolent PCa tissues, 8 aggressive PCa tissues and 6 metastatic PCa tissues. In scenario 4, the 450K methylation profiles of four brain regions of patients with Alzheimer's disease (AD) and normal controls were involved, including the entorhinal cortex (EC) (with 146 AD samples and 97 normal samples extracted from GSE43414 [34], GSE105109 [35] and GSE125895 [36]), the frontal cortex (FC) (with 171 AD and 118 normal samples extracted from GSE43414, GSE66351 [37], and GSE80970 [38], and 308 AD and 233 normal samples extracted from the well-known AD cohort, the Religious Orders Study and Memory and Aging Project (ROSMAP) cohort [39] with Synapase ID syn3157275), and the superior temporal gyrus (STG) (with 135 AD samples and 96 normal samples extracted from GSE43414 and GSE80970), and the hippocampus (HP) (with 17 AD samples and 48 normal samples extracted from GSE125895).

All the 450K datasets were preprocessed with quality control and normalized by using the ChAMP R package [40]. Batch effects were removed by using the Combat function in the sva R package [41], and covariates including sex and age were adjusted using the linear mixed model.

## 3.2 Performance of the DMRIntTk on different tissue pairs

Seven methods were applied to different scenarios to identify DMRs, and the DMR sets predicted on a same dataset were integrated by DMRIntTk. The methylation difference distributions of seven original DMR sets and the integrated one on each pair of tissues are illustrated, as shown in Fig 2. The range of methylation difference [0, 1] is divided into intervals with equal length, as shown in X axis. Y axis denotes the ratio of DMRs with methylation differences falling in each interval to the total number of DMRs in each DMR set. It can be found that the methylation differences of majority of DMRs between a pair of tissues are less than 0.6, and there are large proportions of DMRs with methylation differences less than 0.2. Compared with these original DMR sets, it can be observed that when the methylation difference threshold $t$ is in the range of [0.2, 0.5], the ratios of DMRs in the integrated set are significantly greater than those in the original DMR sets. It suggests that DMRIntTk effectively trim the regions with small methylation differences in original DMR sets by segmenting the genome into bins and weighting the bins, therefore enhances the proportion of DMRs with medium methylation differences.

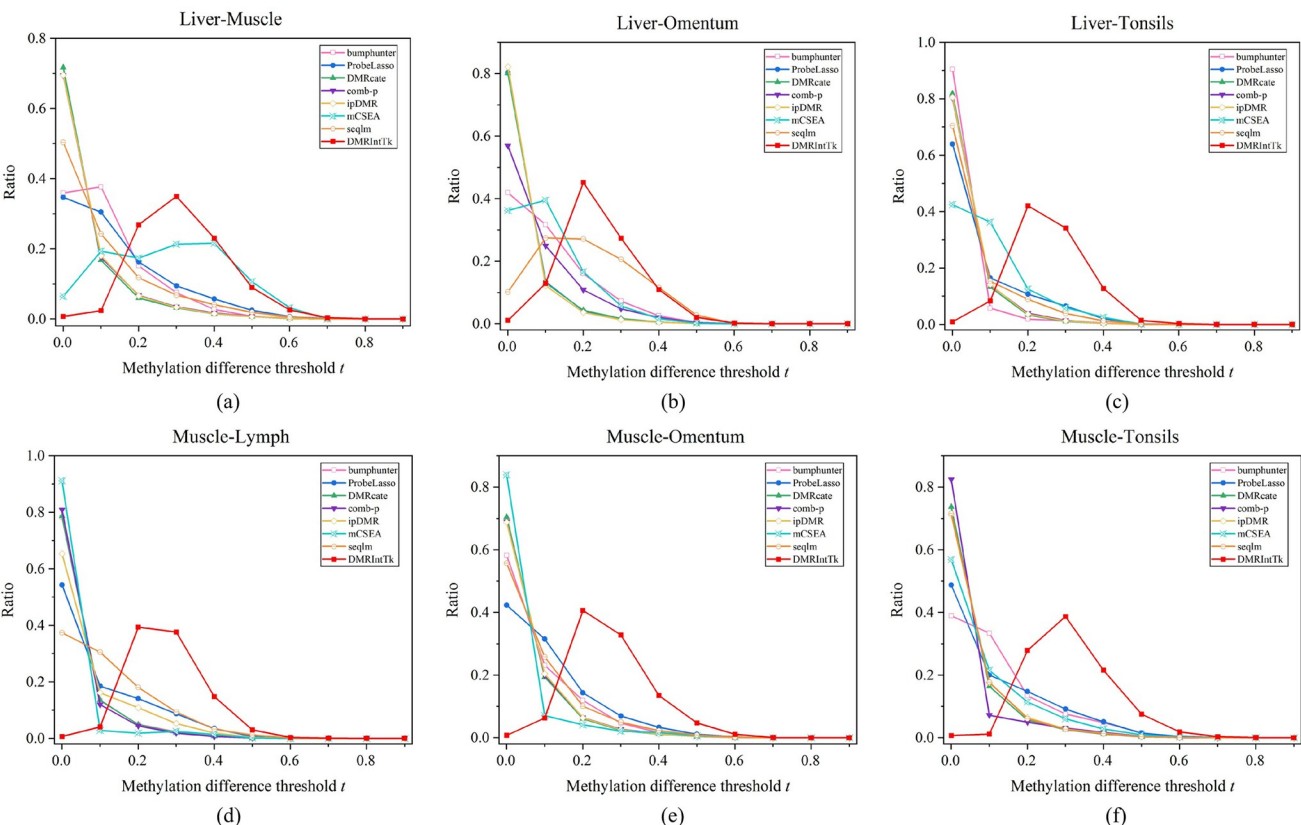

**Fig 2. The methylation difference distributions of DMR sets on six pairs of tissues.** (a) The liver vs. the muscle. (b) The liver vs. the omentum. (c) The liver vs. the tonsils. (d) The muscle vs. the lymph. (e) The muscle vs. the omentum. (f) The muscle vs. the tonsils.

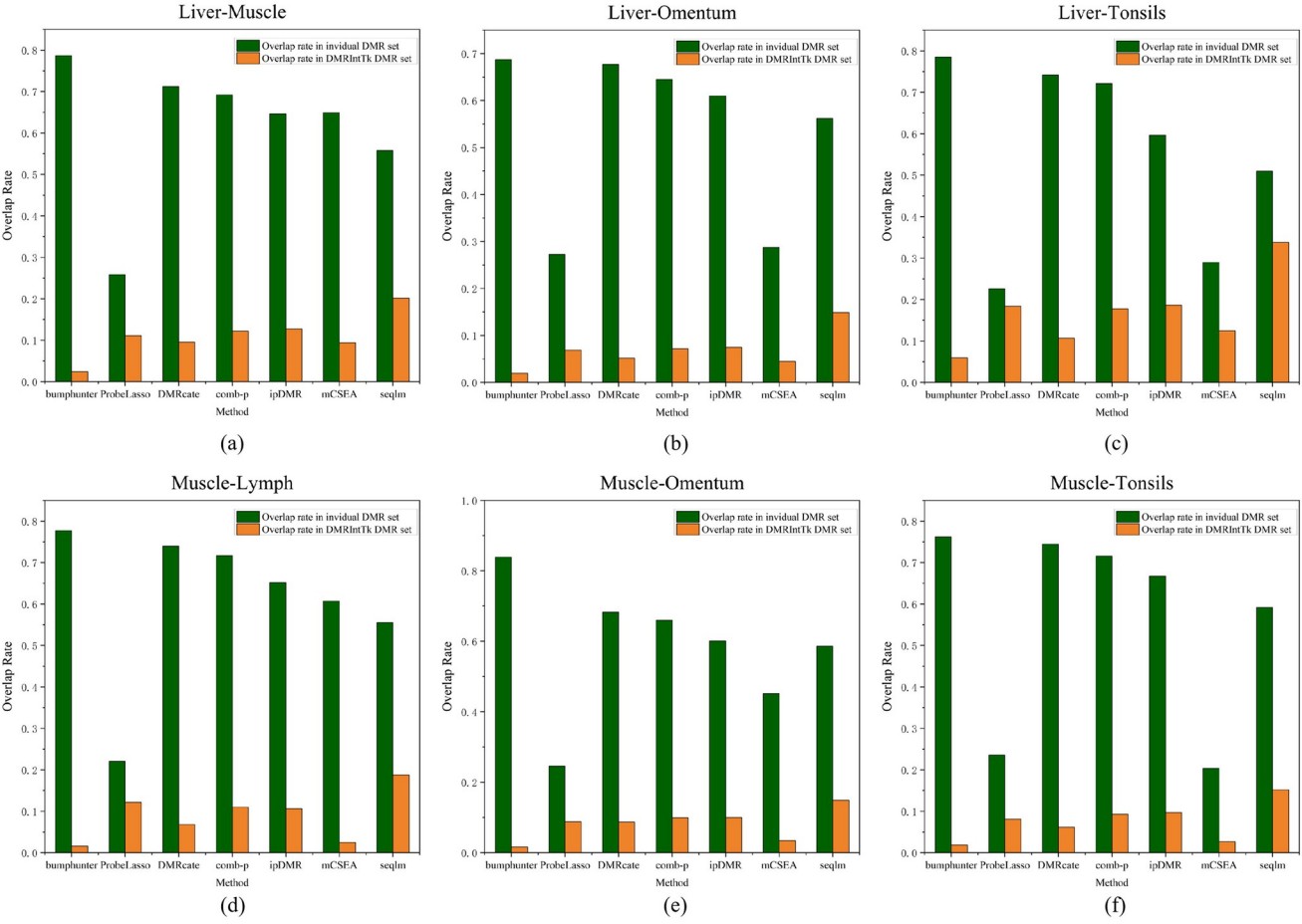

**Fig 3. The overlap rates between the integrated DMR set and individual DMR sets on six pairs of tissues.** (a) The liver vs. the muscle. (b) The liver vs. the omentum. (c) The liver vs. the tonsils. (d) The muscle vs. the lymph. (e) The muscle vs. the omentum. (f) The muscle vs. the tonsils. Overlap rate in individual DMR set denotes the overlap rate calculated against the total lengths of DMRs (with methylation differences $\geq 0.5 * m_{max}$) in an original DMR set, and Overlap rate in DMRIntTk DMR set denotes the overlap rate calculated against the total lengths of DMRs (with methylation differences $\geq 0.5 * m_{max}$) in the integrated DMR set.

To demonstrate that an integrated DMR set is more comprehensive than a single DMR set and contains high reliable DMRs, an overlap analysis was carried between the integrated DMR sets and the original DMR sets. To reduce the bias introduced by DMRs with small methylation differences, only the DMRs with methylation differences not less than $0.5 * m_{max}$ are involved in analysis, where $m_{max}$ denotes the maximum methylation difference in DMRs between two groups. With the overlap length calculated between an integrated DMR set and an original DMR set, two overlap rates are calculated against the total lengths of DMRs (with methylation differences $\geq 0.5 * m_{max}$) in the integrated DMR set and the original DMR set, respectively.

As shown in Fig 3, for each pair of tissues, it can be observed that majority of overlap rates calculated based on the lengths of original DMR sets are greater than 0.6, which indicates that most DMRs with methylation differences $\geq 0.5 * m_{max}$ in these original DMR sets are retained in the integrated DMR set. Further, it can be found out that the overlap rates calculated based on the length of the integrated DMR set are mainly less than 0.2, which indicates the regions with high methylation differences from different DMR sets are effectively integrated and the DMR sets integrated by DMRIntTk are quite comprehensive.

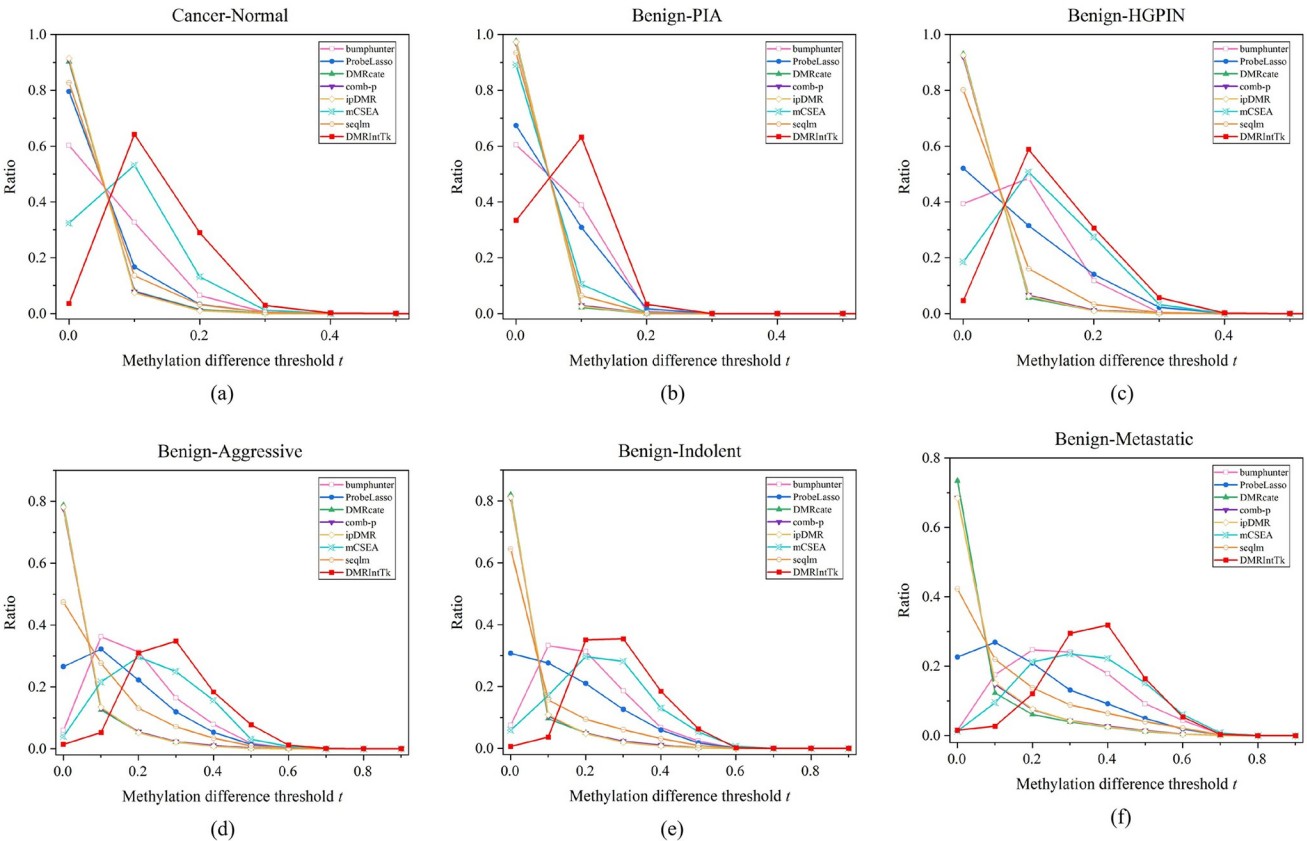

**Fig 4. The methylation difference distributions of DMR sets predicted between PCa and adjacent normal prostate tissues, and between the benign and five other histological stages of PCa.** (a) PCa vs. Normal. (b) Benign vs. PIA. (c) Benign vs. HGPIN. (d) Benign vs. Aggressive. (e) Benign vs. Indolent. (f) Benign vs. Metastatic.

## 3.3 Performance of the DMRIntTk on PCa versus adjacent prostate tissues

The methylation difference distributions of DMR sets between the PCa tissues and adjacent normal tissues are compared, as shown in Fig 4(a). It can be observed that the methylation differences of majority of DMRs between PCa and adjacent prostate tissues are less than 0.3. It can be figured out that when methylation difference *t* is in the range of [0.1, 0.3], the ratios of DMRs in the integrated DMR set are significantly greater than those in the original DMR sets.

The overlap rates between the integrated DMR set and the original DMR sets on the PCa and normal adjacent prostate tissues are compared, as shown in Fig 5(a). It can be found out that the overlap rates calculated against the length of original DMR sets are above 0.4, except for ProbeLasso and mCSEA, while the overlap rates calculated against the integrated DMR set are less than 0.1.

## 3.4 Performance of the DMRIntTk on benign versus other histological stages of PCa tissues

The methylation difference distributions of DMR sets between the benign and five other histological stages of PCa are illustrated, as shown in Fig 4(b)–4(f). It can be found out that, the methylation differences of DMRs identified by senven methods between the Benign and the PIA are less than 0.2, while DMRIntTk enhances the proportion of DMRs with methylation

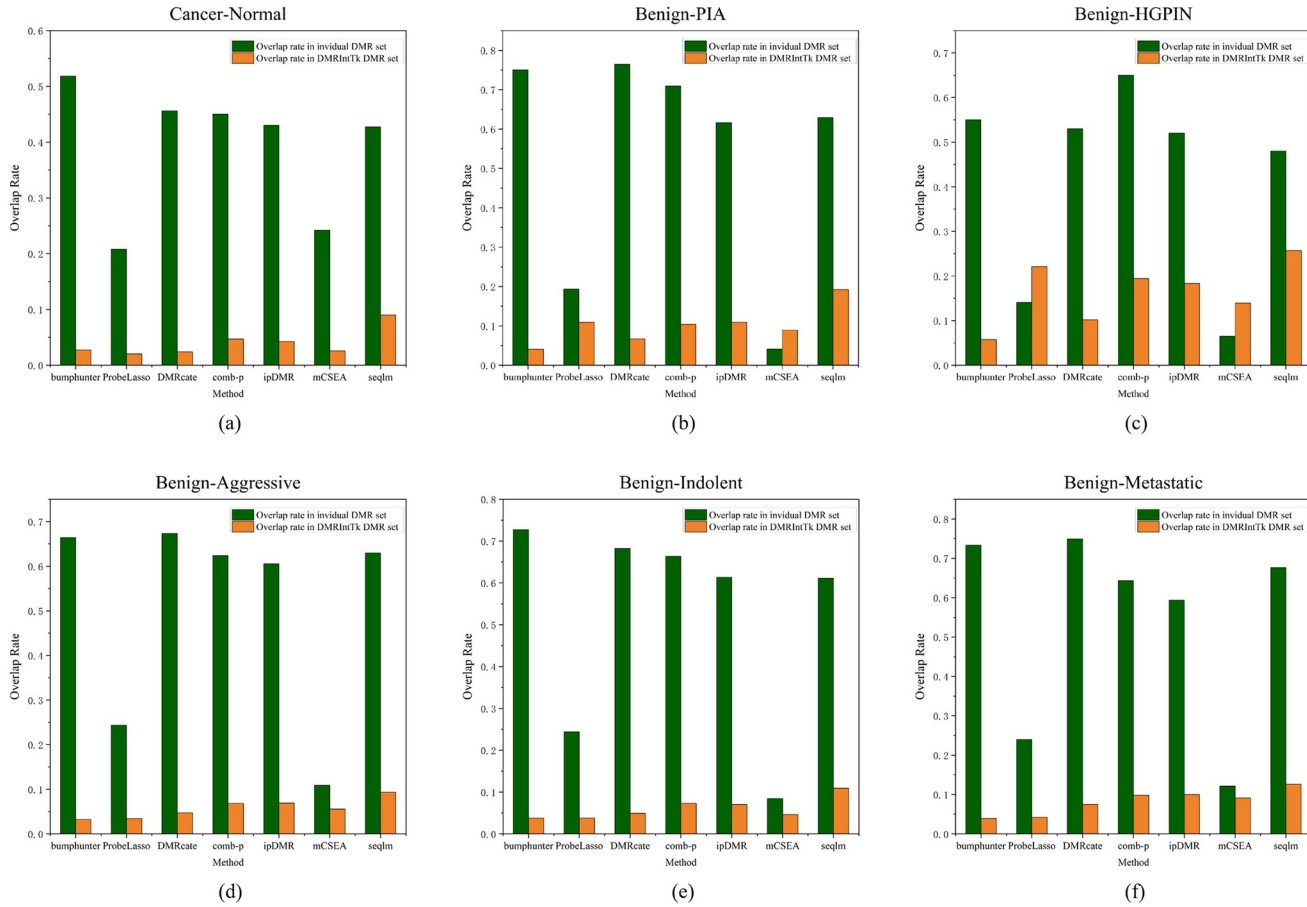

**Fig 5. The overlap rates between the integrated DMR set and individual DMR sets predicted between PCa and adjacent normal prostate tissues, and between the benign and five other histological stages of PCa.** (a) PCa vs. Normal. (b) Benign vs. PIA. (c) Benign vs. HGPIN. (d) Benign vs. Aggressive. (e) Benign vs. Indolent. (f) Benign vs. Metastatic. Overlap rate in individual DMR set denotes the overlap rate calculated against the total lengths of DMRs (with methylation differences $\geq 0.5 * m_{max}$) in an original DMR set, and overlap rate in DMRIntTk DMR set denotes the overlap rate calculated against the total lengths of DMRs (with methylation differences $\geq 0.5 * m_{max}$) in the integrated DMR set.

differences in the range of [0.1, 0.2]. On the Benign vs. the HGPIN, the ratio of DMRs integrated by DMRIntTk is higher than those in the original DMR sets when $t \geq 0.1$. On the Benign vs. the Aggressive, the Benign vs. the Indolent, and the Benign vs. the Metastatic, the ratios of DMRs with methylation differences $\leq 0.1$ predicted by DMRcate, comb-p, seqlm, and ipDMR, are up to 0.8, while bumphunter, mCSEA, and Probelasso predict higher ratios of DMRs with methylation differences in the range of [0.1, 0.4]. On these four pairs, the methylation differences of integrated DMRs are mainly distributed in the range of [0.2, 0.5], and the corresponding ratios are greater than those in the original DMR sets.

For the benign and five other histological stages of PCa, as shown in Fig 5(b)–5(f), we can observe that the overlap rates of most original DMR sets, except for ProbeLasso and mCSEA, are above 0.5 on the benign vs. the HGPIN, and above 0.6 on the benign vs. the PIA, the benign vs. the aggressive, the benign vs. the Indolent, and the benign vs. metastatic. It suggests that nearly half of the DMRs with methylation differences $\geq 0.5 * m_{max}$ in these original DMR sets are included in the integrated DMR set. The overlap rates calculated against the integrated DMR sets are less than 0.2 on the benign vs. the PIA and the benign vs. the HGPIN, and less

than 0.1 on the benign vs. the aggressive, the benign vs. the Indolent, and the benign vs. metastatic. It further validates that the integrated DMR sets are quite comprehensive and very different from the original DMR sets.

It can be observed that the overlap rates between the DMR sets predicted by mCSEA and the integrated DMR sets are low and the ones calculated against the mCSEA DMR set on Benign vs. PIA and Benign vs. HGPIN are higher than those calculated against the integrated DMR sets. The reason is that mCSEA identifies DMRs based on the predefined functional regions, some of which are covered by sparse probes and the bins in these regions are not covered by any probes and the corresponding weights are zeros. Therefore, these bins are not integrated by DMRIntTk.

## 3.5 Performance of the DMRIntTk on the AD versus normal brain tissues

As shown in Fig 6, it can be found that the methylaiton differences between AD patients and normal controls on four brain regions are quite small, less than 0.1. The ratio peaks of most original DMR set are mainly in the range of (0, 0.01], while that of the integrated DMR sets are in the range of [0.02, 0.05]. It can be found out that DMRIntTk gains higher ratios of DMRs with methylation differences in the range of [0.02, 0.04], [0.01, 0.05], [0.02, 0.03] and [0.04, 0.09] than other methods between AD patients and normal controls on the EC, FC, STG and HP, respectively.

For the DMRs predicted from AD patients and normal controls on four brain regions, the overlap analysis between the original DMR sets and the integrated ones is illustrated, as shown in Fig 7. It can be found out that the overlap rates calculated against the original DMR sets are significantly higher than those calculated against the integrated DMR sets, except for the ProbeLasso and seqlm in the EC. It is noteworthy that all DMRs with methylation differences $\geq 0.5 * m_{max}$ in the DMR set predicted by DMRcate in the EC, by bumphunter in the FC, and by mCSEA in the FC and the HP are effectively integrated by DMRIntTK, in which the overlap rates against the original DMR sets are near 1. Since comb-p and DMRcate predict only one DMR with methylation differences $\geq 0.5 * m_{max}$ in the FC and the STG, repetively, the corresponding overlap rates between the DMR sets predicted by comb-p and DMRcate are zero. DMRcate and ipDMR predict none DMRs on the FC, and therefore there are no corresponding overlap rates.

## 3.6 Functional pathway analysis of the integrated DMR sets

To analyze the function of the integrated DMR sets, firstly, the probes contained in each DMR were identified using the annotations for the Illumina methylation arrays. The probes were then mapped to corresponding genes using the annotation to extract the DMR located genes (DMGs). Next, the GO and KEGG enrichment analysis were performed on these DMGs by clusterProfiler R package [42].

To further illustrate the biological advantages of DMRIntTk, we performed the KEGG enrichment analysis comparing the integrated DMR set between PCa tissues and adjacent normal tissues with the original DMR sets.

**3.6.1 GO and KEGG enrichment analysis of the integrated DMR set between PCa tissues and adjacent normal tissues.** The GO enrichment of the integrated DMR set between the PCa tissues and the adjacent normal tissues is illustrated, as shown in Fig 8(a). It can be figured out that the DMGs obtained by DMRIntTk are enriched in biological processes related to cell fate commitment, pattern specification, skeletal system, and axonogenesis.

As we know, epigenetic reprogramming can lead to aberrant lineage specification and transition of tumor cells, which is closely associated with tumor initiation and progression. A

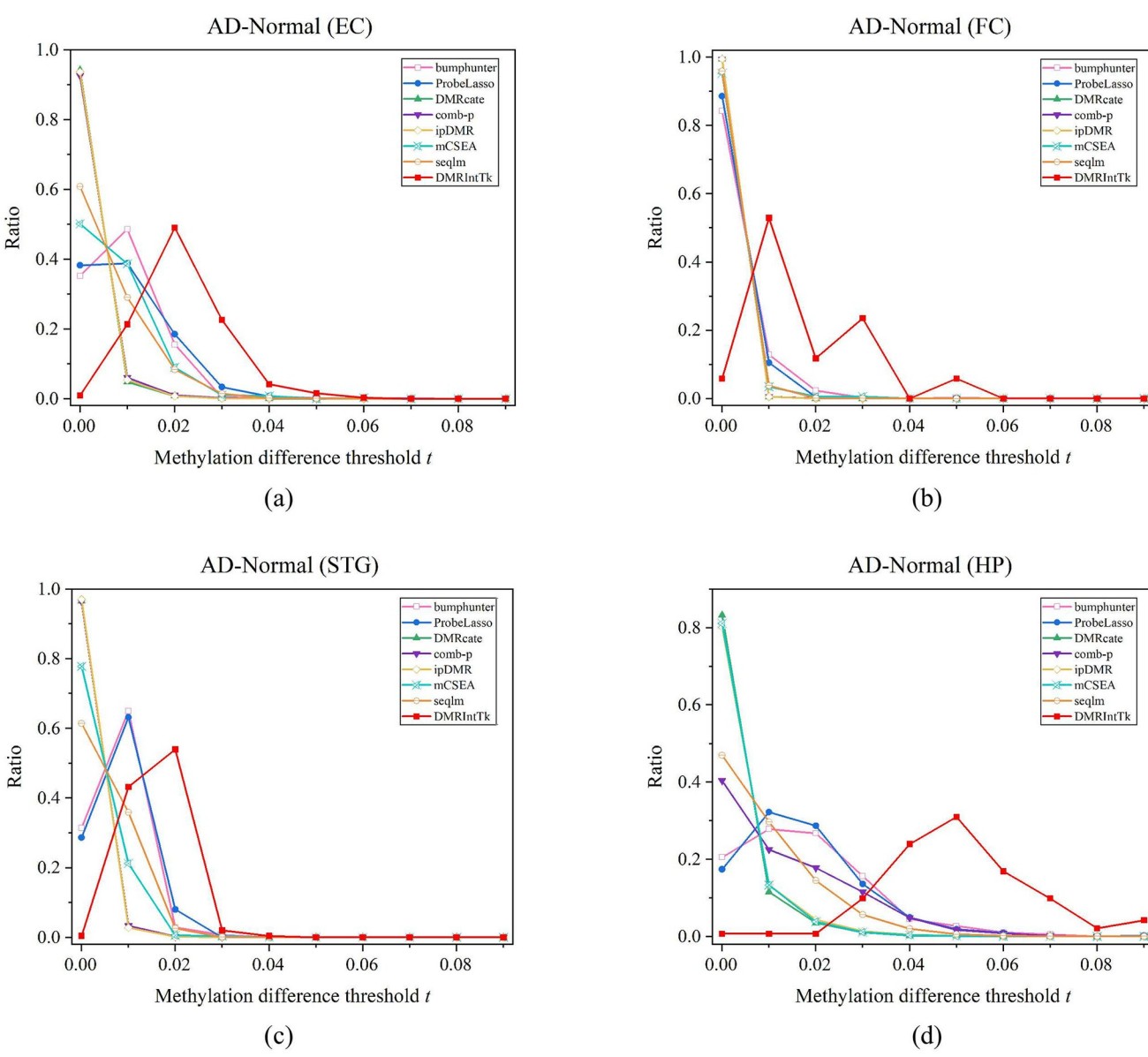

**Fig 6. The distribution of methylation differences of DMR sets between the AD patients and normal controls on four brain regions.** (a) entorhinal cortex (EC), (b) frontal cortex (FC), (c) superior temporal gyrus (STG) and (d) hippocampus (HP).

study [43] showed that overexpression of ERG drives prostate cell fate reprogramming through the orchestration of chromatin interactions, which facilitates the function of ETS transcription factor (ERG) to promote luminal lineage differentiation. Luminal lineage differentiation further leads to luminal cell expansion, which is a structural feature of most PCa compared with normal prostate tissues. Axonogenesis is a biological phenomenon that is crucial in the biology of PCa. Adriana et al. [44] corroborates that axonogenesis is involved in the biological process of the proliferation of PCa through activation of survival pathways and interaction with hormonal regulation. PCa is capable of metastasizing to osteoblasts and inducing extensive new bone deposition. In fact, bone is the most common site of metastasis

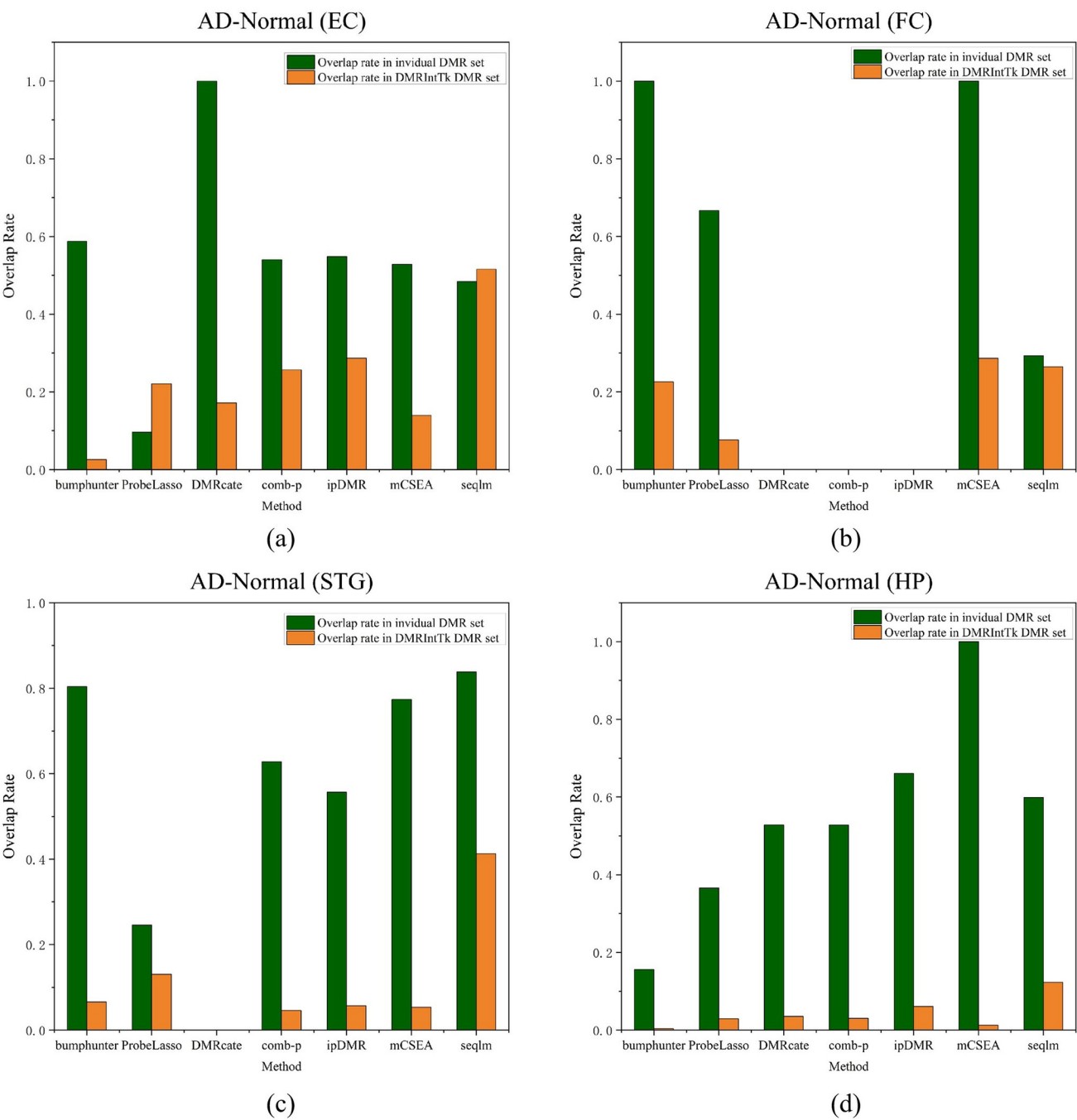

**Fig 7. The overlap rates between the integrated DMR set and individual DMR sets on the four brain regions of the AD patients and normal controls.**
(a) entorhinal cortex (EC), (b) frontal cortex (FC), (c) superior temporal gyrus (STG) and (d) hippocampus (HP).

for advanced solid tumors including PCa [45]. It have been found that approximately 70% of patients with advanced PCa will develop skeletal metastases [46].

The KEGG enrichment analysis, as shown in Fig 8(b), identified neuroactive ligand-receptor interaction as the most significant pathway associated with the integrated DMR set from PCa. Evidence from a recent study [47] indicates a strong link between this pathway and PCa,

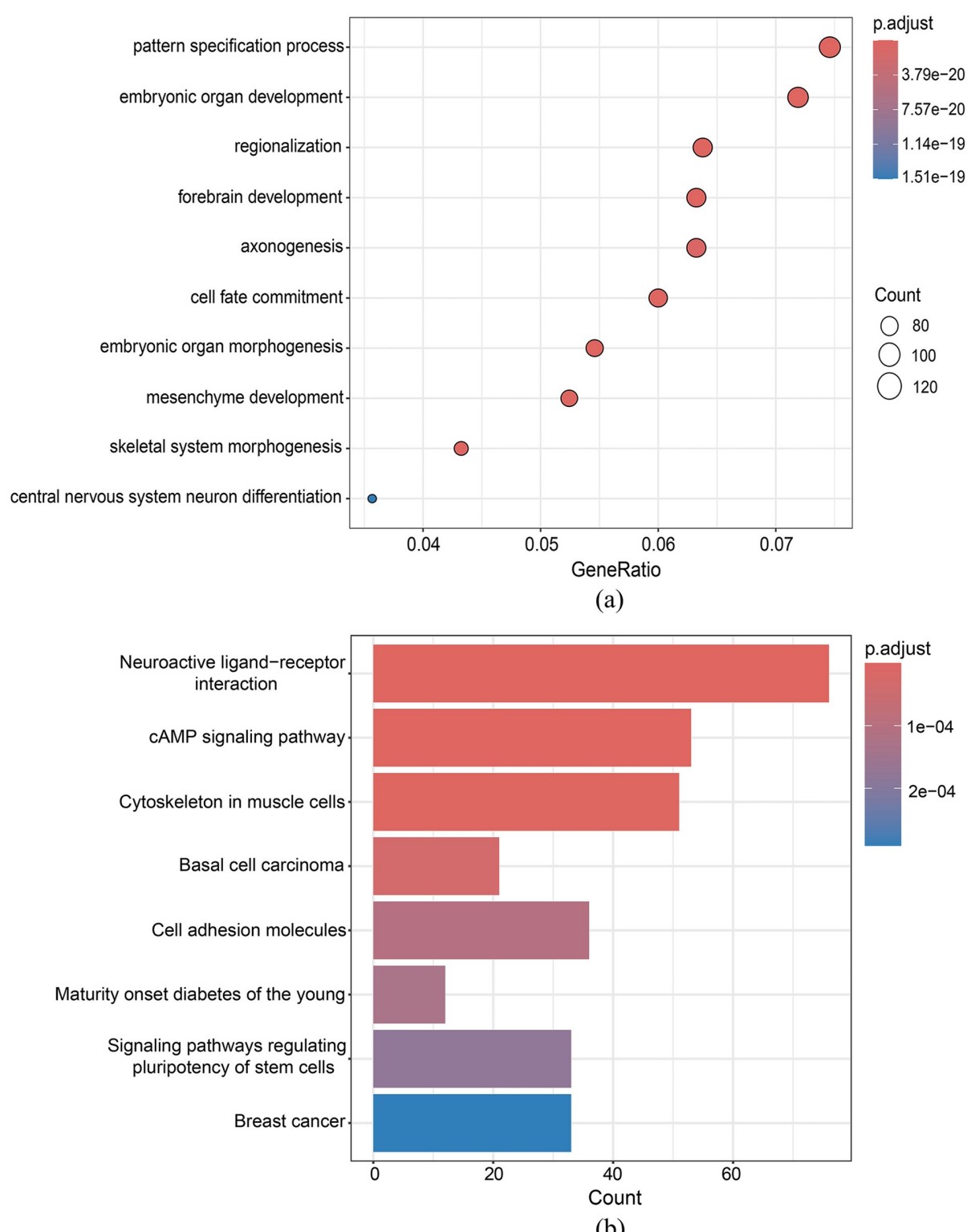

**Fig 8. Functional enrichment analysis of the integrated DMR set on PCa versus normal.** (a) GO enrichment analysis, (b) KEGG pathway enrichment analysis.

particularly in the context of brain metastases. Specifically, the research involving a cohort of 42 patients with PCa brain metastases (PCBM) revealed distinct DNA methylation profiles in PCBM compared with primary PCa. Notably, hypomethylation of promoters for genes associated with neuroactive ligand-receptor interaction, such as GABRB3, CLDN8, and CLDN4, was observed in PCBM, suggesting a specific reprogramming requirement for metastatic cells. This finding underscores the relevance of neuroactive ligand-receptor interactions in the metastatic progression of PCa.

**3.6.2 GO enrichment analysis of the integrated DMR sets between the AD patients and normal controls.** The GO enrichment of the integrated DMR sets predicted between AD patientis and normal controls in the EC brain region is illustrated, as shown in Fig 9(a). It can be observed that the AD-associated DMGs in the EC are mainly enriched in pathways related to cell adhesion, cell nuclear division, and the meiotic cell cycle. Cell adhesion molecules play important roles in the core pathways of AD pathogenesis and progression, including A$\beta$ metabolism, cellular plasticity, neuroinflammation, and vascular changes. For example, expression of neural cell adhesion molecules (NCAM) is considered to be an indicator of neurogenesis, neuronal remodeling and plasticity. In a small study of patients with Parkinson's disease and AD, Guo et al. [48] observed increased levels of soluble NCAM-120 splice variants in the CSF of AD patients compared to controls. Similarly, another study showed a trend towards increased levels of soluble NCAM in the CSF of AD patients compared to controls [49]. The AD-associated DMGs in the EC are also enriched in cell cycle-related pathways. Exposure of the AD brain to extensive stress stimuli may trigger neuronal cell cycle termination. Many studies have shown that cell cycle proteins such as cyclins and CDKs are aberrantly expressed in AD brains [50–52], and CDK inhibitors p16INK4a, p15INK4b, p18INK4c and p19INK4d have also been found to be aberrantly expressed in brain neurons of AD patients [53–55].

The GO enrichment of the integrated DMR sets predicted between AD patientis and normal controls in the STG brain region is illustrated, as shown in Fig 9(b). It can be found out that the AD-associated DMGs between AD patientis and normal controls in the STG are mainly enriched in synapse-related pathways, including glutamatergic synapse, Schaffer collateral-CA1 synapse, modification of synpatic structure and modification of postsynpatic structure. Synapse loss, as a morphological feature that appear early in the onset of AD, has been found to be closely associated with the development of cognitive dysfunction in AD patients. Since calmodulin, particularly neuronal calmodulin (N-cadherins), is important for synapse formation and stability [56], their possible role in AD pathology and clinical disease manifestations has become a hot research topic. Masliah et al. [57] oberserved that the loss of synaptic connections between neurons may facilitate the re-entry of cells into the cell division cycle, resulting in a perturbation of the cell cycle in AD patients. Thus, this available evidence suggests that cell cycle alterations play an crucial role in neurodegeneration in AD.

The AD-associated DMGs between AD patientis and normal controls in the STG are also enriched in spine-related pathways, including dendritic spine and neuron spine. Dendritic spines (DS) are small, highly dynamic prominent structures on the dendritic membrane that form synapses [58]. These structures have specific sub-structural domains with specific functions in synaptic transmission and plasticity and DS are the main sites of structural modification of synaptic plasticity. In neurodegenerative diseases, dynamic morphological changes in the shape and density of dendritic spines can influence their functional features, leading to synaptic dysfunction and cognitive impairment. Peter et al. [59] showed that dendritic spine dysfunction and subsequent synaptic failure are key features of the pathogenesis of AD.

**3.6.3 Comparison of KEGG pathway enrichment analysis between the integrated DMR set and the original DMR sets.** To demonstrate the superiority of DMRIntTk over other

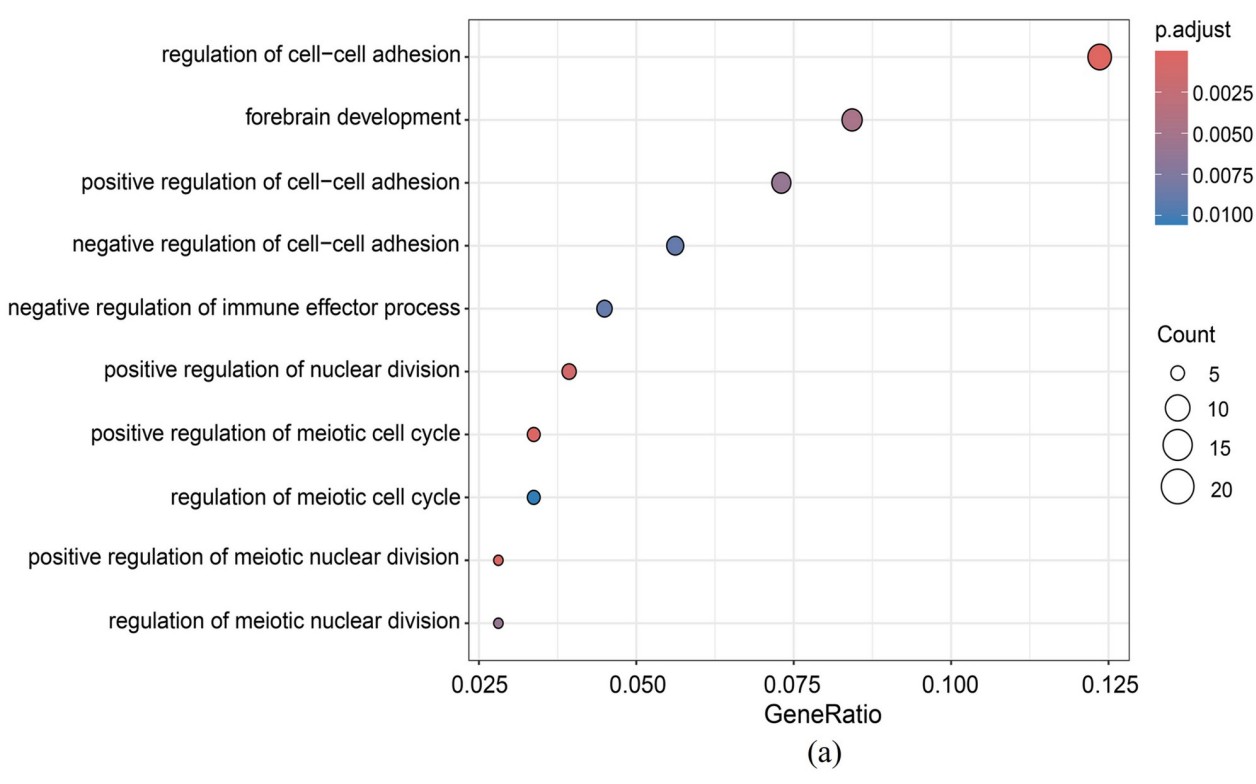

(a)

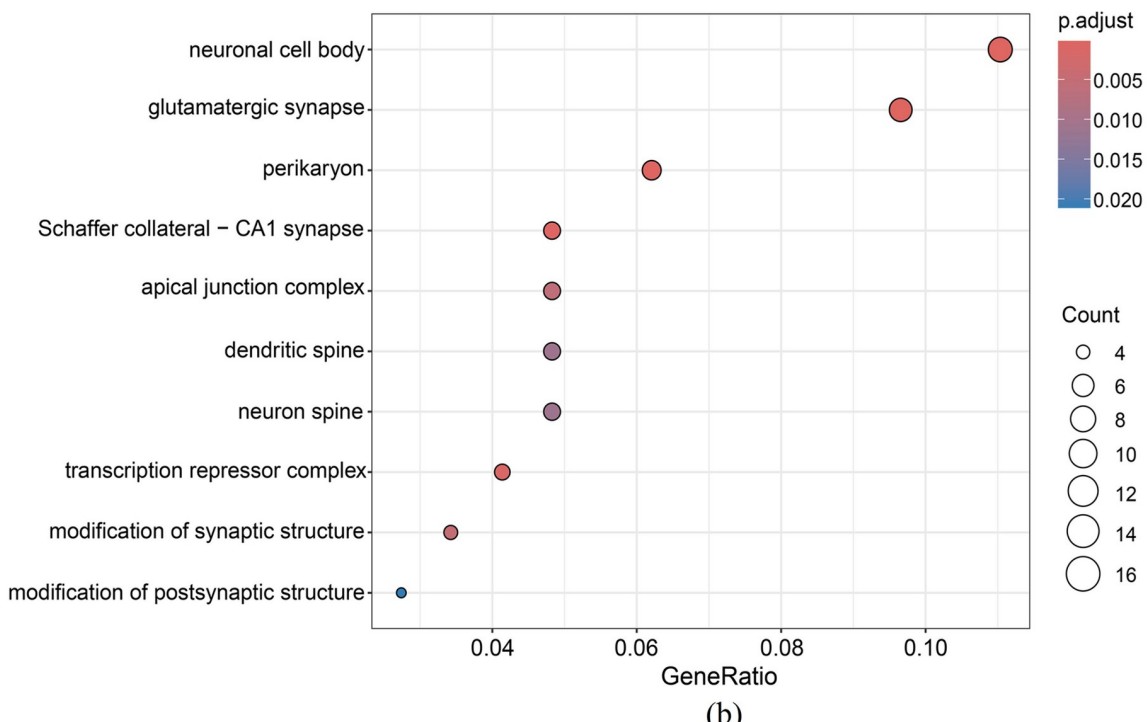

(b)

**Fig 9. GO enrichment analysis of the integrated DMR set on AD versus normal in the (a) entorhinal cortex (EC), and (b) superior temporal gyrus (STG).**

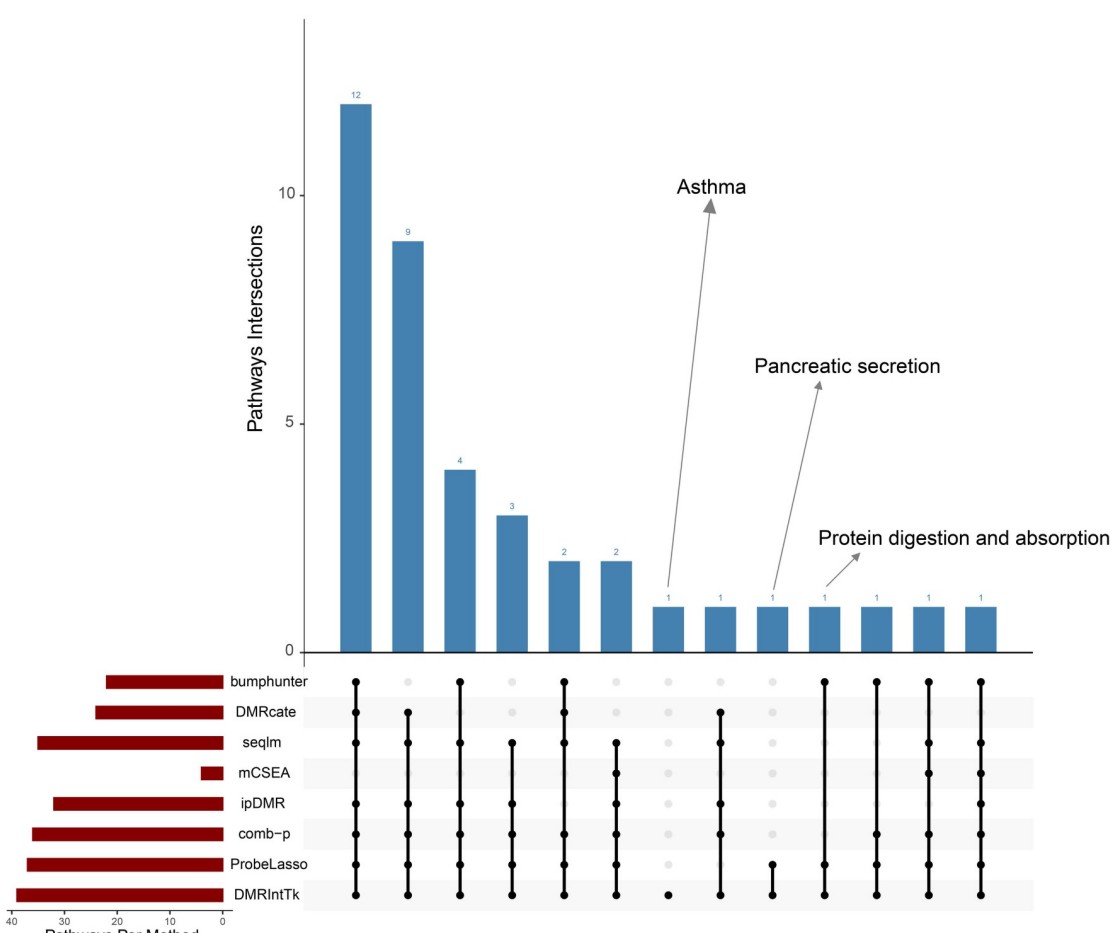

**Fig 10. Upset plot of shared significant pathways between the integrated DMR set and each original DMR set.**

methods, we conducted KEGG pathway enrichment analysis comparing the integrated DMR set between PCa tissues and adjacent normal tissues with the original DMR sets. The number of shared significant pathways between the integrated DMR set and each original DMR set is shown in Fig 10.

The Upset plot illustrates that while most pathways are shared between the integrated and original DMR sets, there are intriguing differences. For instance, the asthma pathway was uniquely significantly enriched in the integrated DMR set and absent from all original DMR sets. Recent studies have explored the association between asthma and PCa, revealing significant insights into potential risk factors [60, 61]. The pancreatic secretory pathway was exclusively enriched in the integrated DMR set and original DMR set predicted by bumphunter, with no enrichment observed in the other six original DMR sets. A insulin receptor-focused study [62] have indicated a potential interplay between pancreatic secretion pathways and PCa malignancy. The protein digestion and absorption pathway was significantly enriched in the integrated DMR set and original DMR sets predicted by bumphunter and ProbeLasso. Raina et al. [63] highlighted the connection between this pathway and PCa, particularly focusing on castration-resistant PCa (CRPC).

Our analysis identified several pathways that were significantly enriched in the integrated DMR set but not in all or the majority of the original DMR sets. Importantly, these pathways have been well-documented in the literature as being associated with PCa, emphasizing the biological relevance and potential insights gained from utilizing the DMRIntTk pipeline. This comparison underscores the unique biological insights provided by our method, which may not be readily apparent when using individual analyses with other methods.

### 3.7 Impact of different genome segmentation strategies on integrated DMRs

To evaluate the impact of different genome segmentation strategies on the integrated DMRs, we have run DMRIntTk with different genome segmentation strategies and compared the integrated DMR sets. Given a length of $k$ bp, first, adjacent probes with a genomic distance of less than k bp were clustered together to form fragments. Then, a $k$ bp sliding window was applied to segment fragments larger than $k$ bp, dividing them into continuous, non-overlapping bins. Specifically, we have applied the lengths of 50 bp, 200 bp, 500 bp, and 800 bp, and the resulting bins were used to integrate the original DMR sets between PCa tissues and adjacent normal tissues using the DMRIntTk pipeline.

As shown in Fig 11, the results indicate that the distributions of methylation differences of integrated DMR sets generated by different segmentation strategies are generally similar. When the methylation difference threshold t falls within the range of [0.1, 0.3], particularly in the range of [0.1, 0.2], we observed that the proportion of DMRs in the integrated set was significantly higher than in the original DMR sets.

Notably, the 50 bp and 800 bp segmentation strategies exhibited a relatively smaller proportion of DMRs with methylation differences greater than 0.2. This could be attributed to the fact that shorter lengths (e.g., 50 bp) may omit informative probes, while longer lengths (e.g., 800 bp) may include regions with minimal methylation differences. Despite these differences, the overall trends were consistent across the different segmentation strategies.

In conclusion, different lengths do have some impact on the integrated DMRs. Based on the comparison of different integrated DMR sets generated by DMRIntTk with different segmentation strategies, we suggest that the length of 500 bp, used in this study, provides a suitable and balanced approach for genome segmentation.

### 4 Conclusions and discussion

By applying DMRIntTk to DMR sets predicted by seven methods across four different scenarios, we demonstrate that DMRIntTk is effective for datasets with varying levels of methylation differences. The methylation difference distributions show that the integrated DMR sets contain a larger proportion of DMRs with higher methylation differences. Furthermore, overlap analysis reveals that the integrated DMR sets include the majority of DMRs predicted by all methods with methylation differences $\geq 0.5 * m_{max}$. Functional analyses, including GO and KEGG pathway enrichment analysis, revealed that the integrated disease-associated DMRs were significantly enriched in biological pathways previously reported to be associated with disease phenotypes. Additionally, KEGG enrichment analysis comparing the integrated DMR set with original DMR sets illustrates that DMRIntTk could capture important pathways that were not captured by all or most of the individual method, emphasizing the biological relevance and potential insights gained from our pipeline. By comparing different integrated DMR sets generated by DMRIntTk with different segmentation strategies, we suggest that the length of 500 bp, used in this study, provides a suitable and balanced approach for genome segmentation.

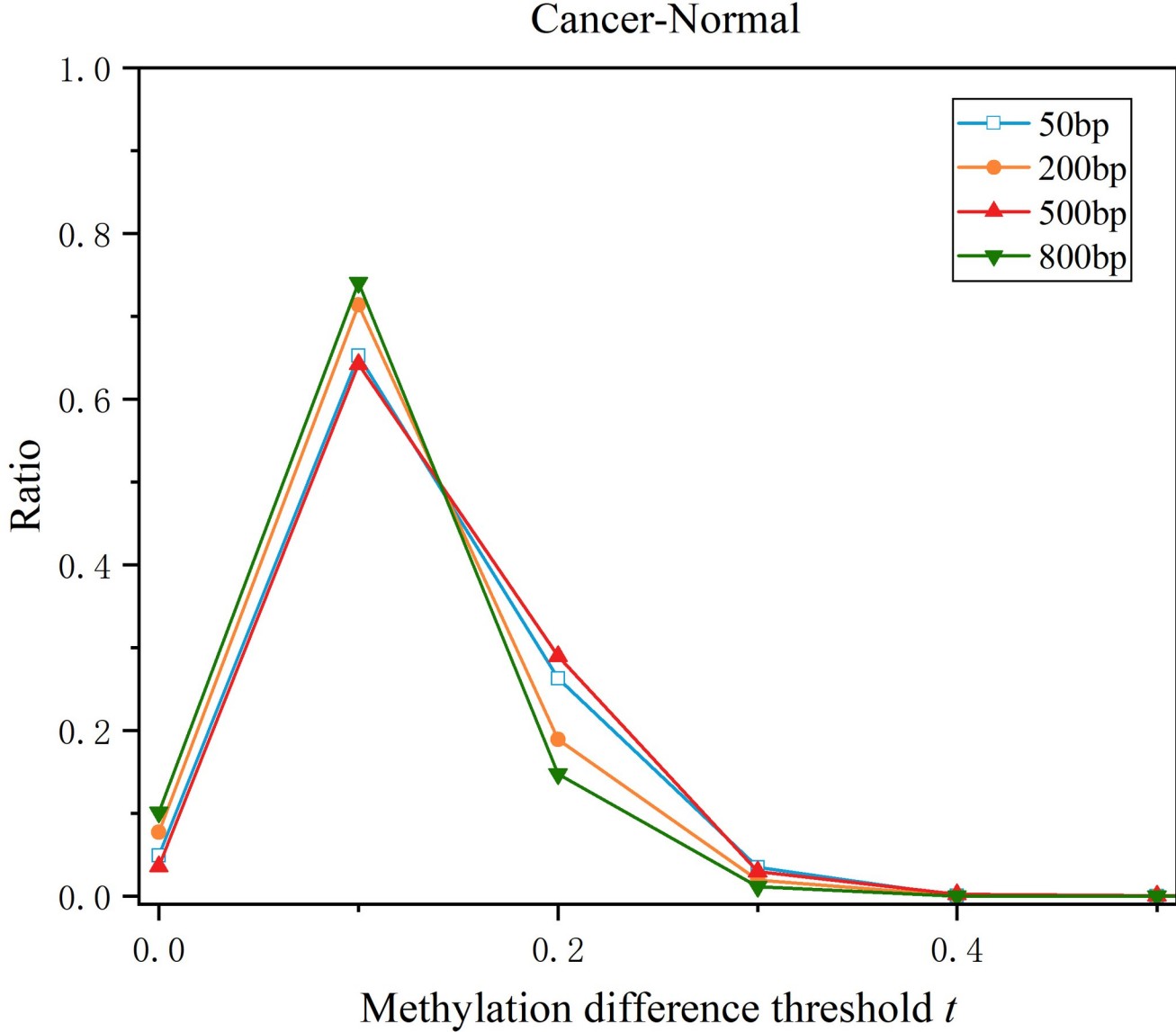

**Fig 11. Methylation difference distributions of different integrated DMR sets generated by DMRIntTk with different segmentation strategies.**

DMRIntTk offers several advantages in integrating DMR sets from different methods. First, it can effectively trim regions with small methylation differences from the original DMR sets by segmenting the genome into bins and weighting these bins based on both methylation differences and the reliability of the covered methods. Second, DMRIntTk is adaptable to datasets with varying methylation differences. This flexibility is achieved through the automatic adjustment of two parameters, $c_t$ and $n_t$, during the integration of bins using the DPC algorithm. These parameters allow clustering thresholds to be tailored to the specific characteristics of each dataset. As a result, DMRIntTk provides a more comprehensive and reliable set of DMRs for downstream analysis, which is crucial for a variety of of applications. Abnormal DNA methylation patterns are closely associated with a wide range of complex diseases, including cancer [64–67], neurological disorders [68–71], and cardiovascular diseases [72, 73]. By

identifying more reliable dysregulated DMRs, DMRIntTk can aid in the discovery of potential biomarkers for early disease detection, prognosis, and treatment response. Furthermore, DMRs are invaluable in tissue origin tracing and liquid biopsy applications. Since DNA methylation landscapes vary across different tissues, specific DMRs can serve as tissue-specific markers [74, 75]. By analyzing methylation patterns in body fluids, the reliable DMR set generated by DMRIntTk can enhance the accuracy of identifying the tissue of origin for tumors or other pathological conditions, which may improve the sensitivity and specificity of diagnostic methods and enable more targeted, personalized treatment strategies and disease progression monitoring.

The current version of DMRIntTk is designed for analyzing DNA methylation data from Illumina methylation arrays, which capture methylation signals across functional regions like CpG islands, promoters, coding regions, and enhancers. While cost-effective, these array platforms are limited by predefined probes, potentially missing complex methylation patterns, particularly in non-coding regions or fragmented DNA including cfDNA, which holds significant potential in non-invasive liquid biopsy applications, including noninvasive cancer screening, non-invasive prenatal testing (NIPT), and the transplantation rejection monitoring and parasitic infection detection [76]. To broaden the applicability for both research and clinical diagnostics, we plan to expand DMRIntTk's capabilities to support data from advanced techniques like bisulfite sequencing (BS-seq), EM-seq, and long-read sequencing. These technologies provide several advantages: BS-seq offers single-base resolution and long-read sequencing allows simultaneous detection of DNA sequence and epigenetic modifications. Future versions of DMRIntTk will incorporate these methods, with adjustments like using smaller genomic segments instead of the 500bp default, given the higher density of cytosines in the genome compared to array probes, and replacing probe counts with the number of methylated cytosines for DMRscore calculation. These changes will enhance the tool's versatility and utility in both research and clinical diagnostics.

Although DMRIntTk was originally developed for analyzing human methylation array data, its underlying methodology and the general framework of the toolkit hold promising potential for expansion to other cases, such as plant systems. DNA methylation regulates gene expression [77], transposon mobility, and genome stability in both plants and animals, though plants exhibit more asymmetric methylation, particularly in CHH and CHG contexts (where H = A, C, or T), as opposed to the symmetric CpG contexts in mammals. Methylation levels in plants vary significantly by species, with *Arabidopsis* showing 24% CpG, 6.7% CHG, and 1.7% CHH methylation [78], while *Betavulgaris* has much higher levels [79] (92.5% CpG, 81.2% CHG, and 18.8% CHH). To adapt DMRIntTk for plants, due to the asymmetric and dispersed methylation patterns in plants, we plan to use adaptive-length bins for genome segmentation rather than the fixed-length bins used in human arrays, so that bin sizes can be adjusted dynamically based on methylation density or variability. Additionally, the abundance of transposons and repetitive sequences in plant genomes calls for incorporating genomic annotations to improve targeted segmentation and analysis. These changes will enhance accuracy and better uncover the links between methylation and gene expression in plants.

## Acknowledgments

We are grateful for resources from the High Performance Computing Center of Central South University.

## Author Contributions

**Conceptualization:** Xiaoqing Peng.

**Methodology:** Xiaoqing Peng.

**Project administration:** Xiaoqing Peng.

**Software:** Wenjin Zhang, Wenlong Jie, Wanxin Cui.

**Supervision:** You Zou, Xiaoqing Peng.

**Validation:** Wenjin Zhang, Guihua Duan.

**Writing – original draft:** Wenjin Zhang.

**Writing – review & editing:** Guihua Duan, You Zou, Xiaoqing Peng.

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
