## [Decision Letter · Decision Letter 0]

18 Sep 2024

PONE-D-24-31358DMRIntTk: integrating different DMR sets based on density peak clusteringPLOS ONE

Dear Dr. Peng,

Thank you for submitting your manuscript to PLOS ONE. After careful consideration, we feel that it has merit but does not fully meet PLOS ONE’s publication criteria as it currently stands. Therefore, we invite you to submit a revised version of the manuscript that addresses the points raised during the review process.

We look forward to receiving your revised manuscript.

Kind regards,

Emidio Albertini, Ph.D.

Academic Editor

PLOS ONE

Journal Requirements: When submitting your revision, we need you to address these additional requirements. 1. Please ensure that your manuscript meets PLOS ONE's style requirements, including those for file naming. The PLOS ONE style templates can be found at https://journals.plos.org/plosone/s/file?id=wjVg/PLOSOne_formatting_sample_main_body.pdf and https://journals.plos.org/plosone/s/file?id=ba62/PLOSOne_formatting_sample_title_authors_affiliations.pdf 2. Please note that PLOS ONE has specific guidelines on code sharing for submissions in which author-generated code underpins the findings in the manuscript. In these cases, we expect all author-generated code to be made available without restrictions upon publication of the work. Please review our guidelines at https://journals.plos.org/plosone/s/materials-and-software-sharing#loc-sharing-code and ensure that your code is shared in a way that follows best practice and facilitates reproducibility and reuse. 3. We note that the grant information you provided in the ‘Funding Information’ and ‘Financial Disclosure’ sections do not match.  When you resubmit, please ensure that you provide the correct grant numbers for the awards you received for your study in the ‘Funding Information’ section. 4. Thank you for stating the following financial disclosure: "This work was supported in part by the Natural Science Foundation of Hunan Province (No. 2022JJ30694 and No. 2022JJ30750); Central South University Innovation-Driven Research Programme (No. 2023CXQD065); Special Funds for Construction of Innovative Provinces in Hunan Province (NO. 2023GK1010)." Please state what role the funders took in the study.  If the funders had no role, please state: ""The funders had no role in study design, data collection and analysis, decision to publish, or preparation of the manuscript."" If this statement is not correct you must amend it as needed. Please include this amended Role of Funder statement in your cover letter; we will change the online submission form on your behalf. 5. Please review your reference list to ensure that it is complete and correct. If you have cited papers that have been retracted, please include the rationale for doing so in the manuscript text, or remove these references and replace them with relevant current references. Any changes to the reference list should be mentioned in the rebuttal letter that accompanies your revised manuscript. If you need to cite a retracted article, indicate the article’s retracted status in the References list and also include a citation and full reference for the retraction notice.

Additional Editor Comments:

Dear authors,

please adjust the manuscript taking into consideration the reviewers suggestions.

Reviewers' comments:

Reviewer's Responses to Questions

**Comments to the Author**

1. Is the manuscript technically sound, and do the data support the conclusions?

Reviewer #1: Yes

Reviewer #2: Partly

Reviewer #3: Yes

2. Has the statistical analysis been performed appropriately and rigorously? 

Reviewer #1: Yes

Reviewer #2: No

Reviewer #3: Yes

3. Have the authors made all data underlying the findings in their manuscript fully available?

Reviewer #1: Yes

Reviewer #2: Yes

Reviewer #3: Yes

4. Is the manuscript presented in an intelligible fashion and written in standard English?

Reviewer #1: Yes

Reviewer #2: Yes

Reviewer #3: Yes

5. Review Comments to the Author

Reviewer #1: The manuscript 'DMRIntTk: integrating different DMR sets based on density peak clustering' by Zhang et al. developed a new toolkit - DRMIntTk - which integrates DMR regions from multiple preexisting software packages to identify DMR regions with high significance in multiple software analysis, but not necessarily only the DRMs with the highest modifications. DRM data from the Illumina 450K chip and submitted previously to NCBI was used for the analysis. The DMRIntTk output (ie Fig 2) does show identification of midrange methylation peaks that are missed by the separate programs based on default settings. Additional tissue studies show a similar enhancement of potential areas of interest using DMRIntTk vs. other software. The author's then use the DRMIntTk data and GO enrichment analysis to show that regions of interest contain genes potentially being influenced via methylation. While the GO enrichment analysis identifies methylated regions of genes that make 'biological sense' for the different tissue analysis done, it is an observation/correlation, not a conclusion of biological function.

Reviewer #2: In this manuscript (PONE-D-24-31358) entitled "DMRIntTk: integrating different DMR sets based on density peak clustering" submitted to PLOS ONE, the authors develop DMRIntTk, a pipeline for the integration of DMRs predicted by other tools. The pipeline was implemented in a four-step method including, genome segmentation, construction of a reliability matrix, bins weighting and DMRs integration. As prove of concept, the authors analyzed different datasets with low, medium and large methylation differences.

The main advantages of DMRIntTk over the other methods are the increased number DMRs with higher methylation differences and the increased proportion of DMRs. Finally, the results of GO enrichment analysis involve terms associated to the samples being analyzed.

Even though the manuscript is well written and clear, my main concern comes from the “biological advantages of the method”. The pipeline seems to have good performance increasing the proportion of DMRs with higher differences, but the question is, it was any important gene/GO term that was not present in the original DMRs? It will be very interesting to see the differences in GO terms and DMGs. On the other hand, to use the pipeline, the users would have to run some or all the other tools before, so it must be clear that the biological results of DMRIntTk are superior to any of the other individual analyses. If the main propose of the pipeline is to find new biological insight that could not be found with other individual tool the authors should present it on the paper. To see a real advantage of DMRIntTk over other methods, more results focusing on the biological aspect are necessary. The authors should amend these issues before publications. See also my comments below:

Introduction:

Please include in the introduction a clear definition of DMRs and the different methods available to detect CpG such as sequencing, array, etc. A paragraph highlighting the main advantage and disadvantage (including references) of the tools would help to understand the strength of the DMRIntTk.

Methods:

Explain better how the bins are obtained. Could the bins introduce a vias in the analysis? Seems that the algorithm was implemented to filter DMRs with low differences. Some levels of low differences are always expected. The different segmentation of the genome could impact on the number of DMRs, was this effect was evaluated?

Please follow the pipeline in the methods sections, move the “Segment the genome and weight the bins” before “Construct reliability matrix”.

Explain clearly the input and output of the pipeline.

Results:

In this section, the authors focus on the ratio of DMRs differences and the overlapping. Please, include the same comparative analysis (i.e, individual tool VS DMRIntTk) with the genes and terms in the different datasets analyzed.

How the DMGs were analyzed? Usually, the gene body methylation is separated from the upstream and downstream methylation.

Discussion:

Discuss what it was stated in the introduction, was DMRIntTk better at finding some type of DMR?

It could be beneficial if the tool could read directly datasets obtained by other methods such as EMseq, bisulfite, long read sequencing etc.

Reviewer #3: In this manuscript, the authors present the results obtained from the integration of different DMR sets and the development a specific toolkit for evaluating the reliability of different DRM sets with the general aim to detect and enhance the proportion of DMR regions with higher levels of methylation.

In general, the paper appears clear in writing. The authors well summarized and presented the importance of their purpose and their case of study. The followed experimental pipeline appears correct and the obtained results have been described quite clearly. However, in Introduction section, there is a little emphasis on the description of the biological process of DNA methylation as epigenetic aspect and the discussion of the results not appears sufficiently clear to define if this new toolkit could be applied also for other cases of study (i.e in plant systems, where the methylation levels related to asymmetric contexts CNN or CNG are frequent). In the absence of a comprehensive description of the biology the final discussion section appears uninvolving. For this reason, it becomes difficult to read and unclear at some points. Formatting the manuscript differently would be important to clearly define some points, for example with a more exhaustive description of the biology behind this system followed by a short section on the technique to take advantage of the biology.

Although the topic is of extreme interest, in my opinion, there are some issues with the focus of the work Therefore, I have described a few major and minor concerns, based on my skills. I had limited my report to some aspects highlighting some issues

Minor comments

Line 58:

“Let mmax denote the maximum methylation difference “ � in my opinion it should be mentioned as it comes measured this value, the authors do it later in the discussion but it should be anticipated.

Equation 1:

Some terms could be explained better (like Ni to the fourth) but maybe I don't have them suitable skills for judging.

Figure 1:

In Figure 1 the expression Diff(bin) is used to indicate the absolute value of the difference of methylation associated with a Bi segment; however, the expression is always used in the text and in the formulas Dif(Bi). In my opinion, if the authors are the same thing, the figure should be corrected, otherwise the figure should be clarified difference to avoid confusion.

Line 365:

A dataset SE50192 is mentioned, it is actually GSE50192 � correct the ID

Line 367:

I was unable to access data from this dataset: Religious Orders Study and Memory and Aging Project (ROSMAP), I don't understand if it's paid or if the type of searching is wrong.

Line 374:

Typo “cluatering” fix clustering

General Personal Opinions:

The part about overlap analysis should be explained better: from the text it is not clear which one is comparing one method at a time with theirs, however, it is clear in the figures.

Moreover, there is a little of confusion with the terms individual DMR set/original DMR set.

As for the validation part of the results, in the end they do a GO for each comparison.

My personal question is related to the possibility to apply an alternative validation analysis to get more evidence.

Finally I personally believe that it is necessary to better explain and evaluate whether and to what extent a similar tool can be used transversally in other biological systems and in other contexts of asymmetric methylation.

6. PLOS authors have the option to publish the peer review history of their article (what does this mean?). If published, this will include your full peer review and any attached files.

Reviewer #1: No

Reviewer #2: No

Reviewer #3: No

---

## [Author Response · Author response to Decision Letter 0]

15 Nov 2024

The point-by-point responses to the kind reviewers and the nice editor are included in the attached file entitled "Response letter".

---

## [Decision Letter · Decision Letter 1]

4 Dec 2024

DMRIntTk: integrating different DMR sets based on density peak clustering

PONE-D-24-31358R1

Dear Dr. Peng,

We’re pleased to inform you that your manuscript has been judged scientifically suitable for publication and will be formally accepted for publication once it meets all outstanding technical requirements.

Kind regards,

Emidio Albertini, Ph.D.

Academic Editor

PLOS ONE

Additional Editor Comments (optional):

Reviewers' comments:

Reviewer's Responses to Questions

**Comments to the Author**

1. If the authors have adequately addressed your comments raised in a previous round of review and you feel that this manuscript is now acceptable for publication, you may indicate that here to bypass the “Comments to the Author” section, enter your conflict of interest statement in the “Confidential to Editor” section, and submit your "Accept" recommendation.

Reviewer #2: All comments have been addressed

2. Is the manuscript technically sound, and do the data support the conclusions?

Reviewer #2: Yes

3. Has the statistical analysis been performed appropriately and rigorously? 

Reviewer #2: Yes

4. Have the authors made all data underlying the findings in their manuscript fully available?

Reviewer #2: Yes

5. Is the manuscript presented in an intelligible fashion and written in standard English?

Reviewer #2: Yes

6. Review Comments to the Author

Reviewer #2: (No Response)

7. PLOS authors have the option to publish the peer review history of their article (what does this mean?). If published, this will include your full peer review and any attached files.

Reviewer #2: No

---

## [Editor Report · Acceptance letter]

8 Dec 2024

PONE-D-24-31358R1 

PLOS ONE

Dear Dr. Peng, 

I'm pleased to inform you that your manuscript has been deemed suitable for publication in PLOS ONE. Congratulations! Your manuscript is now being handed over to our production team.

Kind regards, 

on behalf of

Prof. Emidio Albertini 

Academic Editor

PLOS ONE